# Influence of soot aerosol properties on the counting efficiency of instruments used for the periodic technical inspection of diesel vehicles

Tobias Hammer[1], Diana Roos[1], Barouch Giechaskiel[2], Anastasios Melas[2], Konstantina Vasilatou[1]

[1]Department of Chemistry, Federal Institute of Metrology METAS, Bern-Wabern, 3003, Switzerland

[2] European Commission, Joint Research Centre (JRC), 21027 Ispra, Italy

*Correspondence to*: Konstantina Vasilatou (konstantina.vasilatou@metas.ch)

**Abstract**. In this work, we investigated the influence of different types of soot aerosol on the counting efficiency (CE) of instruments employed for the periodic technical inspection (PTI) of diesel vehicles. Such instruments report particle number (PN) concentration. Combustion aerosols were generated by a prototype bigCAST, a miniCAST 5201 BC, a miniCAST 6204 C and a miniature inverted soot generator (MISG). For comparison purposes, diesel soot was generated by a Euro 5b diesel test vehicle with by-passed diesel particulate filter (DPF). The size-dependent counting efficiency profile of six PN-PTI instruments was determined with each one of the aforementioned test aerosols. The results showed that the type of soot aerosol affected the response of the PN-PTI sensors in an individualised manner. Consequently, it was difficult to identify trends and draw conclusive results about which laboratory-generated soot is the best proxy for diesel soot. Deviations in the counting efficiency remained typically within 0.25 units when using laboratory-generated soot compared to Euro 5b diesel soot of similar mobility diameter (~50-60 nm). Soot with a mobility diameter of ~100 nm generated by the MISG, the lowest size we could achieve, resulted in most cases in similar counting efficiencies as that generated by the different CAST generators at the same particle size, showing that MISG may be a satisfactory - and affordable - option for PN-PTI verification; however, further optimization will be needed for low-cost soot generators to comply with European PN-PTI verification requirements.

## 1 Introduction

Soot particles emitted by transport sources can have adverse health effects (Kheirbek et al., 2016; US-EPA, 2019; WHO, 2021). To reduce particulate emissions, new procedures for the periodic technical inspection (PTI) of diesel vehicles based on the measurement of particle number (PN) concentration have recently been established in Switzerland, Germany, the Netherlands and Belgium, while other countries might follow in due time (EU, 2023; Vasilatou et al., 2022). Portable instruments known as PN-PTI counters are used for measuring particle number concentration (PNC) directly in the tailpipe of diesel vehicles equipped with a diesel particle filter (DPF) (Kesselmeier and Staudt, 1999; Melas et al., 2021, 2022, 2023). When the DPF is intact, the emitted PNC is low (typically up to a few thousand particles per $cm^3$), whereas if the DPF is defect or tampered, PNC increases to several hundred thousand particles per $cm^3$ (Botero et al., 2023; Burtscher et al., 2019; Giechaskiel et al., 2022). In terms of particle mass concentration, a functioning DPF can reduce particulate emissions by up to a factor of 150 (Ligterink, 2018) while in terms of particle number concentration a solid particle number trapping efficiency of higher than 99 % has been reported in the literature (Frank, Adam et al., 2020). It has been shown that a small

fraction (about 10 %) of vehicles with defective DPF is responsible for up to 80-90 % of the total fleet emissions (Burtscher et al., 2019; Kurniawan and Schmidt-Ott, 2006). The goal of PN-PTI procedures is to identify diesel vehicles with compromised DPFs, thus ensuring that vehicles in operation maintain their performance as guaranteed by type-approval, without excessive degradation, throughout their lifetime (EU, 2023).

Although the concept of PN-PTI is simple, its implementation in practice is not as straightforward. PTI procedures are not fully harmonised and, as a result, the limit values for the emitted PNC, the technical specifications of the PN-PTI counters and the test protocol for type-examination and verification are defined at a national level (Anon, n.d.; AU-Richtlinie, n.d.; PTB, 2021; UVEK, 2023; VAMV, 2018; Vasilatou et al., 2022, 2023). Differences in national legislations might lead to contradicting results, e.g. the same diesel vehicle might pass the PTI check in one country but fail in another one. To ensure fair implementation of regulations across Europe and avoid unnecessary costs which may occur for vehicle owners after a False Fail, the various PTI procedures must be compared and the differences elucidated.

PN-PTI instruments go through a type-examination procedure which may differ in each country. Among several tests, type-examination includes a counting efficiency and a linearity check typically performed with combustion aerosols. During their lifetime, PN-PTI instruments are checked for their linearity with polydisperse particles (typically with a GMD of 70 ± 20 nm). In our previous study (Vasilatou et al., 2023), we showed that the choice of test aerosol during type-examination or verification of PN-PTI instruments significantly affects the performance of instruments based on diffusion charging (DC). When sodium chloride (NaCl) or carbonaceous particles from spark-discharge generators were used as test aerosols, the counting efficiency of the DC-based instruments changed by up to a factor of two compared to that exhibited with diesel soot. The experiments clearly showed that soot from laboratory-based combustion generators was the best proxy for soot emitted by diesel engines, however, potential differences between the different combustion generators available on the market were not investigated.

In this study, we challenged six different DC-based PN-PTI instruments with polydisperse soot particles produced by three different CAST generators (Jing AG, Switzerland), the miniature inverted soot generator (MISG, Argonaut Scientific, Canada) and a Euro 5b diesel vehicle. The geometric mean diameter of the test aerosol was in the range used for linearity checks of PN-PTI instruments as well as in typical size range emitted by diesel engines. The scope of our study was to investigate possible differences that may arise when using different combustion aerosol generators during the type-examination and verification of PN-PTI instruments as well as to correlate with diesel engine emitted soot. We focused on DC-based instruments because we expect a larger impact of the aerosol properties on their response compared to CPC-based ones (Vasilatou et al., 2023). The size-dependent counting efficiency of the PN-PTI instruments was determined by using a condensation particle counter (NPET 3795, TSI Inc., USA) as a reference instrument. We discuss the results in the context of the different national legislations and make recommendations for the harmonisation of the various calibration and verification procedures in the laboratory.

## 2 Materials and methods

During the first measurement campaign at METAS, the following laboratory-based diffusion or premixed flame generators were used to produce test aerosols: a prototype bigCAST, a miniCAST 5201 BC (Ess et al., 2021b; Ess and Vasilatou, 2019), a miniCAST 6204 C and the miniature inverted soot generator (MISG) (Giechaskiel and Melas, 2022; Kazemimanesh et al., 2019; Moallemi et al., 2019; Senaratne et al., 2023). By varying the operation

points of the CAST generators, polydisperse aerosols with a geometric mean mobility diameter ($GMD_{mob}$) ranging
from 50 nm to 100 nm were generated, as summarised in Fig. S1. In the case of the MISG, particles with a $GMD_{mob}$
down to 100 nm were produced in a repeatable and stable manner using a mixture of dimethyl ether and propane
(Senaratne et al., 2023). This is in agreement with another study, where the modal diameter varied between 95 and
158 nm (Bischof et al., 2020).
The counting efficiency profiles (CE) of six DC-based PN-PTI counters, namely the AEM (TEN, the Netherlands),
HEPaC (developed by the University of Applied Sciences Northwestern Switzerland and distributed by Naneos
GmbH, Switzerland), DiTEST (AVL DiTEST, Austria), CAP3070 (Capelec, France), DX280 (Continental
Aftermarket & Services GmbH, Germany) and AIP PDC KG4 (referred to as Knestel hereafter, KNESTEL
Technologie & Elektronik GmbH, Germany) were determined experimentally. The HEPaC, DiTEST, CAP3070
and DX280 had been type-approved at METAS according to the Swiss regulations (VAMV, 2018) whereas the
Knestel instrument had been type-approved according to the German regulation (AU-Richtlinie, n.d.). The
experimental setup at METAS is depicted in Fig. 1a. Soot produced by CAST-burners or the MISG was passed
through a catalytic stripper (CS, Catalytic Instruments GmbH, Germany), a Nafion dryer (MD-700-12S-1,
PERMA PURE, U.S.A.), a VKL 10 diluter (Palas GmbH, Germany) and a custom-made dilution bridge, and was
mixed and diluted with filtered air in a 27-ml-volume chamber. To deliver the aerosol into the mixing volume, a
blower (Micronel AG, Switzerland) was used. The aerosol was split with a custom-made 8-port flow splitter and
delivered simultaneously to the devices under test (DUT, in this case PN-PTI instrument) and the reference particle
counter (NPET 3795, TSI Inc., USA). The splitter bias was determined according to the procedure specified in the
ISO 27891 standard and was found to be within 1 % for particles with a $GMD_{mob}$ equal to or larger than 23 nm. In
addition, the length of the tubes from the flow splitter to the devices was adapted to the respective flow rate to
ensure equal diffusion losses. The NPET was selected as reference instrument for two reasons; i) it could be used
in field measurements as it included a dilution system, a volatile particle remover and a particle counter, ii) during
type examination portable PN-PTI instruments are typically used as reference. NPET had been calibrated in a
traceable manner according to the ISO 27891 standard, and showed a CE of $0.58 \pm 0.02$, $0.77 \pm 0.02$, $0.77 \pm 0.01$,
$0.80 \pm 0.01$ and $0.79 \pm 0.02$ at a $GMD_{mob}$ of 23 nm, 50 nm, 70 nm, 80 nm and 100 nm, respectively and this
counting efficiency was taken into account during data analysis (i.e. calibration factors in the range 1.72 - 1.28
were applied to the concentrations reported by the NPET depending on the particle size).

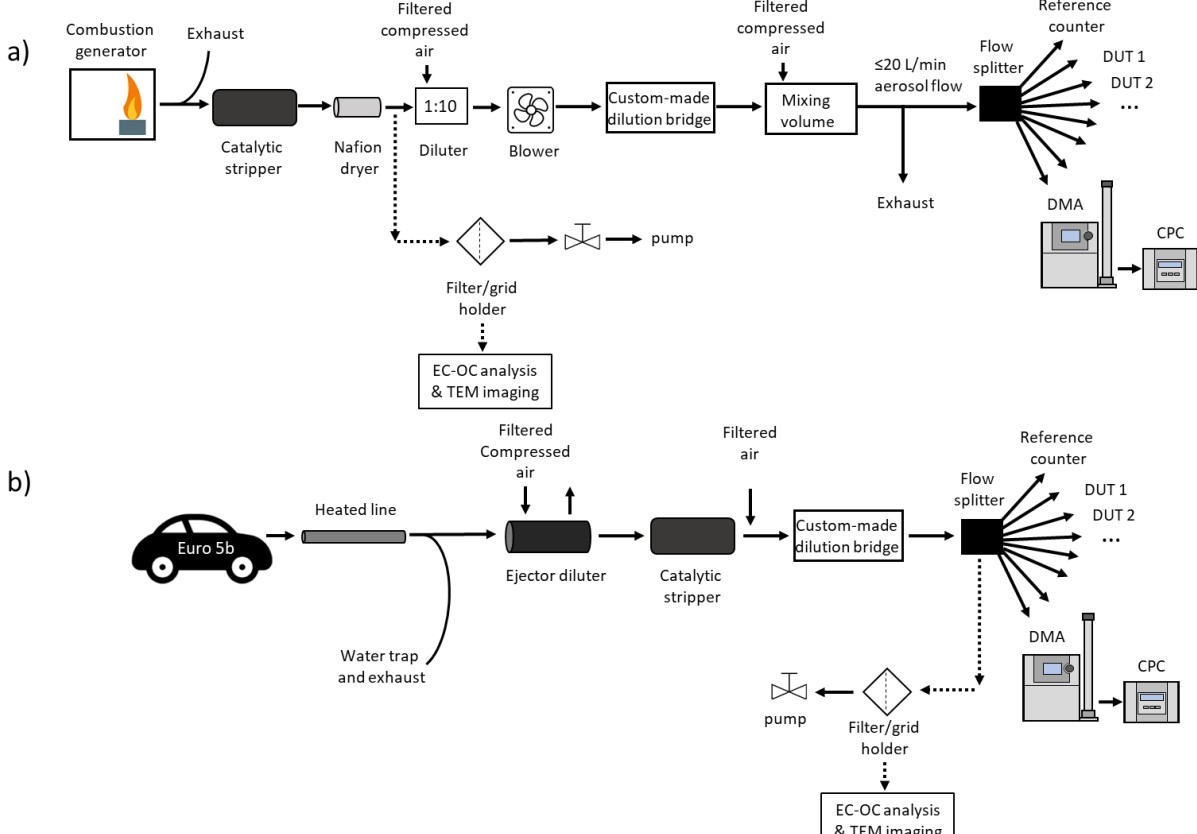

**Figure 1: a)** Experimental setup for the verification of PN-PTI instruments in the laboratory. Four different combustion generators were used (see text for more details). DUT stands for device under test. Dashed arrows designate measurements which were performed separately, i.e. not in parallel with PN-PTI verification. **b)** Experimental setup as used for field measurements at JRC.

Mobility size distributions were recorded simultaneously by a scanning mobility particle sizer ([85]Kr source 3077A, DMA 3081 and butanol CPC 3776, TSI Inc., USA). To analyse the morphology of the soot particles, particles were sampled for 5 s with a flow rate of 1.2 L/min downstream the Nafion dryer, collected on copper-coated TEM (transmission electron microscopy) grids placed in a mini particle sampler (MPS, Ecomeasure, France) and analysed with a Spirit Transmission EM (Tecnai, FEI Company, USA). Soot particles were also sampled on QR-100 Advantec filters (Toyo Roshi Kaisha, Ltd. Japan, preheated at 500 °C for > 1 h) for durations of 15 – 30 min. Elemental carbon (EC) to total carbon (TC) mass fractions were measured with an OC/EC Model 5L analyser (Sunset Laboratory Inc., NL) by applying an extended EUSAAR-2 protocol (Ess et al., 2021b, 2021a). In a second measurement campaign at JRC, the HEPaC, DiTEST, CAP3070 and DX280 counters were challenged with real diesel engine exhaust from a Euro 5b vehicle. Fig. 1b depicts the experimental setup at JRC. Soot from engine exhaust was passed through a water trap, a heated line (150 °C) to avoid water condensation, an ejector dilutor (DI-1000, Dekati, Finland), a catalytic stripper (Catalytic Instruments GmbH, Germany) to remove (semi)volatile organic matter, and was diluted to the required concentrations with a custom-made dilution bridge. It has been shown that the ejector dilutor does not affect the particle size distribution (Giechaskiel et al., 2009). PNC was recorded for several minutes, which allowed identifying long-time trends or drifts of the reported PNC. In addition, PNCs were averaged over a period of 1 min, thus the duration was similar to the duration of real PN-PTI tests which varies from 15 to 90 s. Mobility size distributions were measured by an SMPS, consisting of an [85]Kr source (3077A, TSI Inc., U.S.A.; purchased in 2021), a DMA 3081 and a CPC 3010 (TSI Inc., USA).

A Euro 5b vehicle with by-passed DPF was tested as real source of diesel soot. The vehicle generated size
distributions with a $GMD_{mob}$ of 56.4 nm $\pm 0.7$ nm. Diesel particles from the Euro 5b vehicle were collected on
TEM grids and quartz filters and analysed as described above.
The fractal dimension $D_f$ of size-selected soot particles with a mobility diameter $d_p$ of 100 nm was derived via
image analysis of high-quality TEM-images using the FracLac feature of ImageJ 1.53e (ImageJ, National institutes
of Health, USA). In a first step, the greyscale TEM-images were converted into binary images utilizing the auto-
convert function of FracLac. In a second step, the $D_f$ values were determined via the so-called box counting,
averaging 12 rotations of each image. The effective density was determined for the 100 nm setpoints using an
Aerodynamic Aerosol Classifier (AAC, Cambustion, UK) and a DMA (TSI Inc., USA) in tandem as described in
(Tavakoli and Olfert, 2014).
**3 Results**
**3.1 Aerosol properties**
Particle number concentration measured by diffusion chargers depends on the average number of charges carried
by each particle (Fierz et al., 2011). Particle size and morphology have been shown to have an effect on the number
of charges carried by the particles and, thus, on the counting efficiency of diffusion charger based PN-PTI
instruments (see (Dhaniyala et al., 2011; Vasilatou et al., 2023) and references therein). Soot particles form
complex structures described by a fractal-like scaling law (Mandelbrot, 1982), and their mobility is influenced by
their morphology (described by the fractal dimension and fractal pre-factor) and the momentum-transfer regime
(Filippov et al., 2000; Melas et al., 2014; Sorensen, 2011). To characterise the soot particles produced by the
different aerosol generators, the following aerosol properties were determined: particle size distribution, EC/TC
ratio, primary particle size and fractal dimension. EC/TC ratio can also have an effect on the morphology of the
soot particles. Soot particles formed in premixed flames (i.e. high EC/TC) exhibit a loose agglomerate structure
where the primary particles are clearly distinguishable from one another, while soot generated in fuel-rich flames
(high OC/TC) has a more compact structure and the primary particles tend to merge with each other (see Fig. 3 in
(Ess et al., 2021b)). OC stands for organic carbon.

The properties of the soot aerosols are summarised in Table 1. Mobility size distributions and TEM images are
shown in Fig. S1 and Fig. 2, respectively.
**Table 1: Physical properties of the soot aerosols produced by the various combustion generators and the Euro 5b engine**.
$GMD_{mob}$ and GSD stand for geometric mean mobility diameter and geometric standard deviation. EC and TC denote elemental
and total carbon. $d_{pp}, \rho_{eff}$ and $D_f$ are the primary particle diameter, effective density and fractal dimension of soot particles.

| Soot generator | Setpoint | $GMD_{mob}$ (nm) | GSD (nm) | EC/TC mass fraction (%)* | $d_{pp}$ (nm)** | $\rho_{eff}$ (g/cm³)*** | $D_f$[††] |
|---|---|---|---|---|---|---|---|
| MISG | 100 nm | 103.3 | 1.76 | $86.2 \pm 10$ | $9.2 \pm 2.8$ | $0.91 \pm 0.02$ | $1.63 \pm 0.08$ |
| miniCAST 6204 C | 50 nm | 50.7 | 1.43 | $57.2 \pm 8.9$ | | | |
| | 70 nm | 73.4 | 1.48 | $27.9 \pm 4.6$ | | | |
| | 80 nm | 80.0 | 1.54 | $77.8 \pm 9.0$ | | | |
| | 100 nm | 99.5 | 1.69 | $41.9 \pm 6.5$ | $21.6 \pm 2.5$ | $0.35 \pm 0.04$ | $1.64 \pm 0.09$ |

| | | | | | | | |
|---|---|---|---|---|---|---|---|
| miniCAST 5201 BC | 50 nm | 51.1 | 1.60 | 100 ± 18.5 | | | |
| | 70 nm fuel-lean | 75.3 | 1.59 | 94.6 ± 15.6 | | | |
| | 70 nm fuel-rich | 74.2 | 1.69 | 73.7 ± 11.4 | | | |
| | 80 nm | 81.8 | 1.57 | 98.1 ± 15.3 | | | |
| | 100 nm fuel-lean | 99.8 | 1.63 | 97.4 ± 9.6 | 15.8 ± 3.5[†] | ~ 0.4[†] | 1.55 ± 0.11 |
| | 100 nm fuel-rich | 101.9 | 1.58 | 65.7 ± 10.0 | Primary particles are partly merged[†] | 1.04 ± 0.16[†] | 1.65 ± 0.08 |
| bigCAST | 50 nm | 52.5 | 1.57 | 50.9 ± 11.7 | | | |
| | 70 nm | 71.6 | 1.54 | 62.2 ± 13.3 | | | |
| | 80 nm | 81.5 | 1.53 | 81.2 ± 8.8 | | | |
| | 100 nm | 98.9 | 1.60 | 100.0 ± 9.0 | 24.5 ± 1.8 | 0.66 ± 0.04 | 1.57 ± 0.05 |
| Vehicle Euro 5b | | 56.4 | 2.12 ± 0.00 | 83.5 ± 20.5 | 19.7 ± 4.4 | | |

* Uncertainties of the EC/TC mass fraction (downstream of the CS) are estimated to be in the range of 10-15 %.
Uncertainties due to the split point could not be quantified and were not taken into account.
** Expanded uncertainty ($k$ =2, 95 % confidence interval) determined as the twofold standard deviation of $d_{pp}$, of
at least 20 primary particles of various mature soot particles divided by the square route of the number of
measurements.
*** Expanded uncertainty ($k$ =2, 95 % confidence interval) determined as the twofold standard deviation of three
measurements.
† Taken from (Ess et al., 2021b).
†† Expanded uncertainty ($k$ =2, 95 % confidence interval) determined as the twofold standard deviation of at least
10 measurements.

The $D_f$ values summarised in Table 1 represent the average values obtained from at least 20 particles for each type
of soot. These values agree well with those reported in previous studies for bare (i.e. freshly emitted) soot particles
(Pang et al., 2022; Wang et al., 2017).
. The lowest effective density (0.35 ± 0.02 g/cm$^3$) was found for particles generated by the miniCAST 6204 C.
Considering that these particles contain a high amount of OC, this value might seem at first glance to be low, but
can be explained by the highly fractal-like structure of soot (Fig. 2e). In comparison, the miniCAST 5201 BC
produced particles with an effective density of 1.04 ± 0.08 g/cm$^3$ when operated under fuel-rich conditions (i.e.
high OC mass fraction), which is in line with the more compact structure as shown in (Ess et al., 2021b). Similarly,
the MISG generated particles with an effective density of 0.91 ± 0.02 g/cm$^3$. 100 nm particles generated by the
bigCAST exhibited an intermediate effective density of 0.66 ± 0.02 g/cm$^3$. According to the summary work by
Olfert and Rogak, the effective density of denuded soot from various sources (gas turbines, compression ignition
engines and laboratory-based burners) lies typically in the range 0.4-0.8 g/cm$^3$ at 100 nm mobility diameter (Olfert
and Rogak, 2019). Compression-ignition engines tend to produce soot with higher effective densities, while gas-
turbine soot tends to have lower effective densities (Olfert and Rogak, 2019). The calculated fractal dimension of
soot particles lied in the range 1.55 – 1.65 for all generators, in line with the fractal-like morphology observed in
the TEM images and with previous studies on freshly emitted soot particles from different combustion sources
(Pang et al., 2023)..

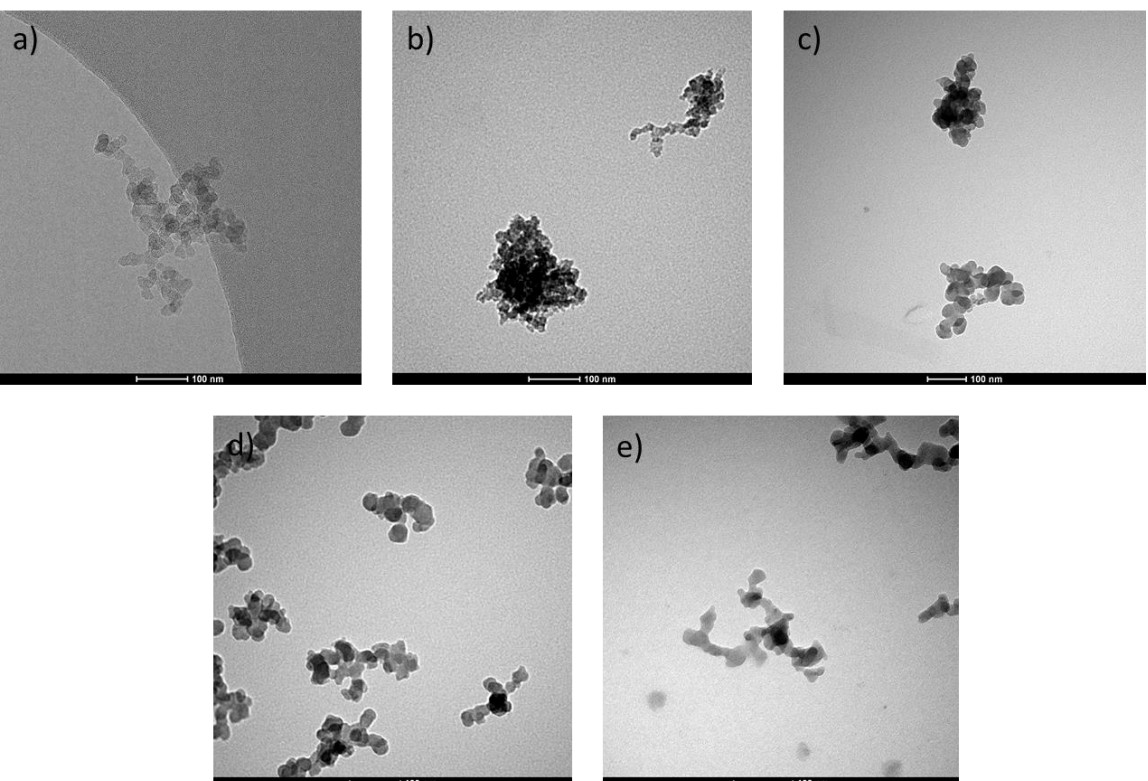

**Figure 2: TEM images of polydisperse soot particles generated by a) the miniCAST 5201 BC (GMD$_{mob}$ of ~100 nm, fuel-**
**lean setpoint); b) the MISG (GMD$_{mob}$ of ~100 nm); c) by the Euro 5b test vehicle (GMD$_{mob}$ of ~55 nm); d) the prototype**
**bigCAST (GMD$_{mob}$ of ~100 nm); and e) by the miniCAST 6204 C (GMD$_{mob}$ of ~100 nm). Further images are compiled**
**in Figs. S2-S5 and in** (Ess et al., 2021b)**.**
Soot particles generated by the bigCAST with a GMD$_{mob}$ of ~ 100 nm consist of primary particles with a diameter
$d_{pp}$ = 24.5 nm ± 1.8 nm, whereas those from miniCAST 5201 BC (fuel lean setpoint) have an average primary
particle size of 12.3 nm ± 3.7 nm at a similar GMD$_{mob}$. Soot generated by the MISG had a much smaller primary
particle size ($d_{pp}$ of 9.2 nm ± 3.8 nm). The TEM images in Figs. 2b and S3 revealed that some particles have a
more compact soot structure than what reported by (Kazemimanesh et al., 2019) who used ethylene as fuel. This
observation is in line with the relatively high particle effective density (0.91 g/cm$^3$) reported above.
**3.2 Counting efficiency (CE) profiles of PN-PTI counters**
The CE profiles of the PN-PTI instruments under test were determined by dividing the reported number
concentration by that measured with a reference condensation particle counter (NPET 3795, TSI Inc., USA). The
counting efficiency of the reference counter was taken into account during the data analysis.
Figure 3 summarises the results obtained with the various laboratory-based combustion generators and the Euro
5b diesel vehicle. In general, the CE of PN-PTI instruments increased with increasing GMD$_{mob}$, in line with
previous studies (Melas et al., 2023; Vasilatou et al., 2023). In the case of CAP3070, CE started to decrease at
GMD$_{mob}$ ≥ 65 nm, most probably due to built-in correction factors. It cannot be ruled out that the measurement
principle of the instrument, based on the so-called escaping current principle, plays also a role (Lehtimäki, 1983).
In general, for each PN-PTI instrument, the differences in CE when challenged with different soot aerosols of
similar particle size were <0.25 at 50 nm and increased with size, but remained typically lower than 0.5. Higher
differences were observed for CAP3070 at around 100 nm, probably related to the internal correction factors. This
indicates that the exact morphology (e.g. primary particle size, effective density) of the test aerosol had an effect
on instrument performance as expected from previous studies (Dhaniyala et al., 2011). The response of each PN-
PTI model was, however, individual, making it difficult to draw any general trends. For instance, the CE of the
HEPaC was higher when measuring soot particles from the miniCAST 6204 C compared to soot of similar
$GMD_{mob}$ from the bigCAST. CAP3070 showed the opposite behaviour. At a $GMD_{mob}$ of ~100 nm, DX280
exhibited a higher CE with soot particles generated by the miniCAST 5201 BC under fuel-rich conditions (i.e.
lower EC/TC mass fraction) than at fuel-lean conditions (higher EC/TC mass fraction). CAP3070 showed again
the opposite behaviour. It is also worth mentioning that for the HePAC and DX280 instruments the measured CE
values scattered more at particle sizes larger than 90 nm. This supports the choice of soot with 50-90 nm mobility
diameter for the PN-PTI instruments verification linearity tests. The counting efficiency of the different PN-PTI
counters as a function of time is shown in Figs. S6-S9 for a measurement duration of 2 min.

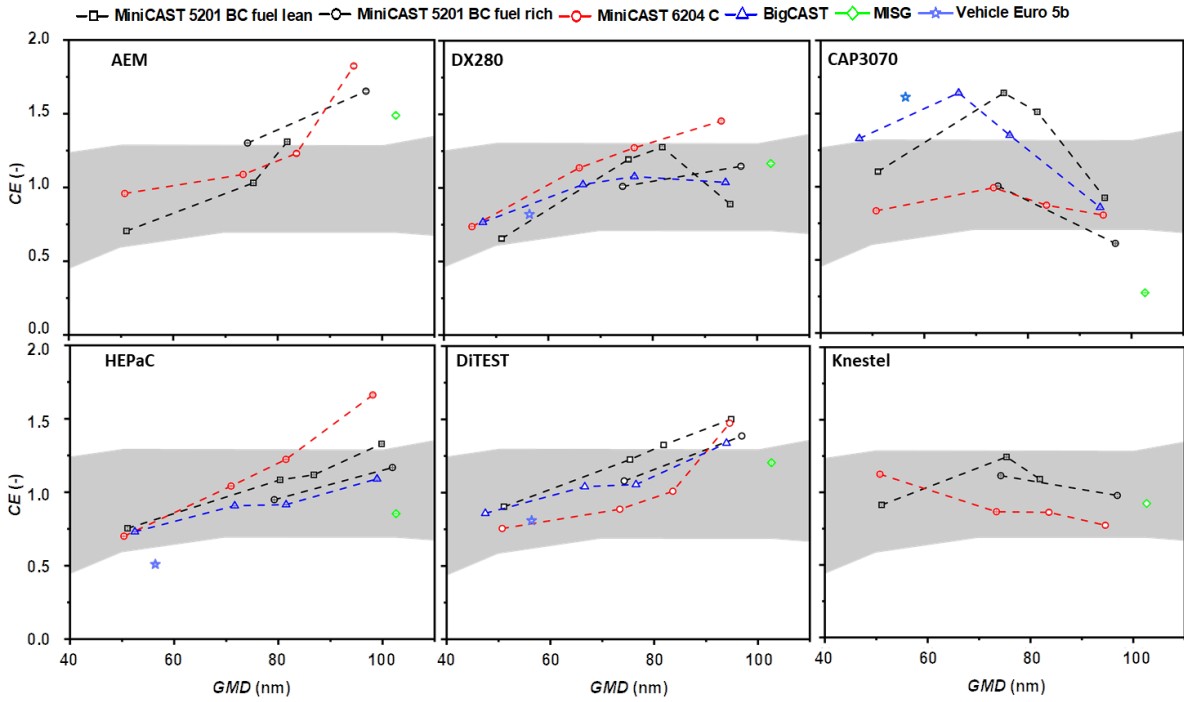


**Figure 3: Influence of the type of soot generator/vehicle engine (bigCAST, miniCAST 5201 BC, miniCAST 6204 C, MISG and Euro 5b diesel engine) on the counting efficiency (CE) of six different PN-PTI counters: AEM, HEPaC, DiTEST, CAP3070, DX280, and Knestel. The grey-shaded area designates the upper and lower limits in the counting efficiency as defined in the document "Commission Recommendation on particle number measurement for the periodic technical inspection of vehicles equipped with compression ignition engines" (EU, 2023).**

In the case of the DX280 and DiTEST, the CEs reported for the laboratory-generated soot ($GMD_{mob}$ of about 50-
55 nm) showed an excellent agreement with the CE measured for diesel soot from a Euro 5b vehicle as shown in
Fig. 4a. In all other cases, deviations were observed. These remained typically within 0.25 units in CE but in one
case (for CAP3070) reached a factor of 2. Note that for real vehicle exhaust the tolerance (maximum permissible
error MPE) according to German regulations is ± 50% (PTB, 2021). In general, the data indicate that soot produced

by miniCAST and bigCAST generators simulate, in most cases, the properties of diesel soot by a Euro 5b vehicle satisfactorily.

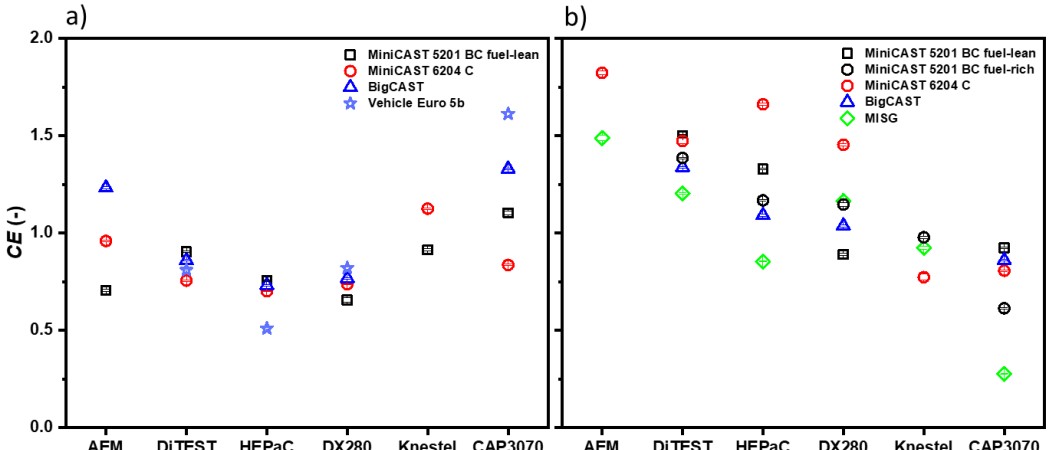

**Figure 4: Influence of the type of soot generator/engine (bigCAST, miniCAST 5201 BC, miniCAST 6204 C, MISG, Euro 5b vehicle) on the counting efficiencies (CE) of six different PN-PTI counters: AEM, HEPaC, DiTEST, CAP3070, DX280, Knestel (the Knestel and AEM counters were not challenged with the Euro 5b vehicle since the Knestel counter was sent for service and the performance of the AEM counter deteriorated during the measurement campaign at JRC). The polydisperse test aerosols had a particle number concentration of ~100'000 cm$^{-3}$ and a GMD$_{mob}$ of a) 50-55 nm and b) ~ 100 nm.**

As shown in Fig 4b, soot generated by the MISG (GMD$_{mob}$ ~ 100 nm) led to CEs close to 1 for the DX280, DiTEST, Knestel and HEPaC counters, and the CEs lied within the tolerance range defined in Germany and Switzerland (the Netherlands and Belgium only specify a tolerance range for mobility diameters up to 80 nm). The CE limit values were only exceeded in the case of the AEM and CAP3070 counters but this was most probably due to a deterioration of the performance of the AEM instrument or an underestimated internal correction and an overestimated internal correction factor in the case of CAP3070. Although the size of the soot generated by the MISG (GMD$_{mob}$ ≥90 nm) tends to be larger than real soot from diesel engines (Kazemimanesh et al., 2019; Moallemi et al., 2019; Senaratne et al., 2023), it's ease of operation combined with the affordable price make it an attractive choice for PN-PTI verification in the laboratory.

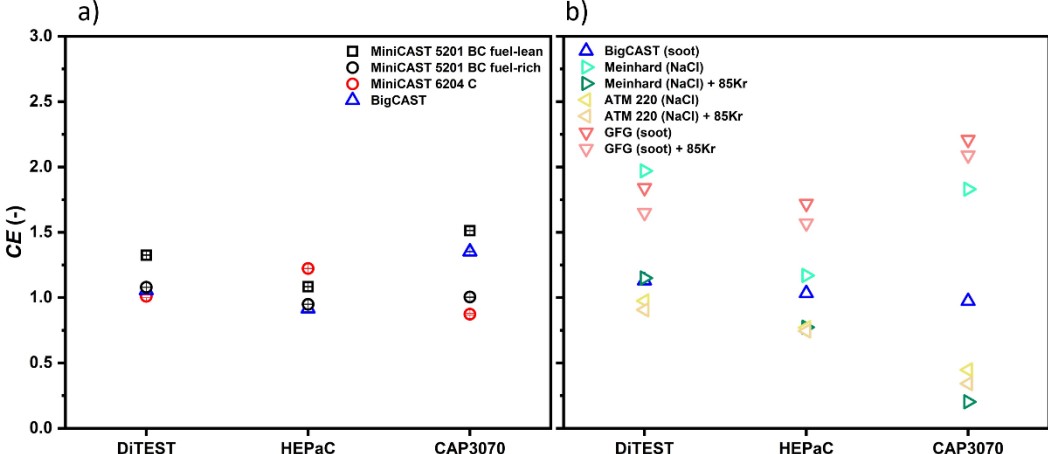

**Figure 5: a) Influence of different soot aerosols with a GMD$_{mob}$ of ~80 nm on the counting efficiencies (CE) of three different PN-PTI counters. b) Influence of different test aerosols (soot, NaCl and carbonaceous particles from a spark-**

**discharge generator) on the counting efficiencies (CE) of the same PN-PTI counters. The test aerosols had a GMD$_{mob}$**

**255 of ~80 nm. The data points are taken from (Vasilatou et al., 2023).**

The variation in the counting efficiency of the PN-PTI instruments when tested with soot particles from different
combustion generators (Fig. 5a) is much smaller than that observed with test aerosols such as NaCl or particles
from a spark-discharge generator with a similar GMD$_{mob}$ (Fig. 5b) (Vasilatou et al., 2023). For instance,
carbonaceous particles from a GFG spark-discharge generator (Palas GmbH, Germany) led to a CE of ≥2 in the
case of CAP3070 and 1.7-1.8 in the case of DiTEST. On the contrary, CE remained typically in the range 0.7-1.3
when soot was used as test aerosol, irrespective of the type of combustion generator (Fig. 5a). Further studies with
more diesel test vehicles would be necessary to elucidate which type of laboratory-generated soot is the best proxy
for diesel soot, keeping in mind that the properties of real diesel soot can also differ considerably, depending on
the engine design, driving cycle and fuel properties (Hays et al., 2017; Wihersaari et al., 2020).

**265 4 Recommendations**

Based on the results of this and previous studies (Vasilatou et al., 2023), the following recommendations can be
made:
1) Initial and follow-up verification of DC-based PN-PTI counters should ideally be performed with soot as test
aerosol. If possible, the same type of combustion generator should be used for the determination of CE during
type-examination and verification.
2) Low-cost soot generators can be a stable source of combustion particles and can be employed for PN-PTI
verification using the appropriate setup correction factors. However, the GMD they produce should be in the
range 70±20 nm in order to comply with the current linearity verification requirements in Europe.
3) Laboratory procedures for PN-PTI type-examination and verification should be further harmonised in Europe
to avoid inconsistencies in the enforcement of PTI legislation. International round robin tests should be
performed to examine whether a) the various PN-PTI instruments type-examined and verified in different
European countries according to national regulations exhibit a similar performance and b) whether PN-PTI
instruments verified in the same country but with different test aerosols identify defect DPFs in a consistent
manner.
As highlighted in our previous study (Vasilatou et al., 2023), "setup correction factors" should be determined
whenever verification is performed with particles other than soot to account for the effects of the test aerosol on
the instrument's counting efficiency. These "setup correction factors" depend on both the aerosol physicochemical
properties and the instrument's design, and need to be determined at the NMI level at regular intervals as drifts in
the performance of the aerosol generator may occur. If "setup correction factors" are not applied or are inaccurate,
the reliability of PTI will be compromised. The use of "setup correction factors" is more critical when nebulisers
or spark-discharge generators are used, but special care should also be given to different flame soot generators.
This calls for a closer collaboration between NMIs, state authorities, instrument manufacturers and verification
centres to ensure fair implementation of regulations in Europe. Further harmonisation of the different PN-PTI
type-examination procedures in Europe, e.g. in terms of the combustion generator, would be a valuable first step
in order to determine meaningful correction factors for other test aerosols.

**5 Conclusions**

The type of soot aerosol generated by diffusion and premixed flame generators affected the response of six different DC-based PN-PTI counters tested in this study. Size and physicochemical properties of the test aerosol had effects on the CE of all counters, but the effect was different for each counter. In most cases, the different laboratory-generated soot aerosols resulted in deviations of 0.25 units in the counting efficiency of individual counters compared to Euro 5b diesel soot at similar mobility diameters (~50-60 nm). It is not entirely clear which type of laboratory-generated soot is the best proxy for real soot emitted by diesel vehicles as the response of the PN-PTI instruments to the different test aerosols was not uniform. It must also be kept in mind that the properties of diesel soot may vary depending on the engine specification and operation. Nevertheless, the differences observed with different soot generators were much lower compared to previous studies that used NaCl and particles from spark discharge generators. This study confirms that soot aerosols, irrespective of the generator model, are more suitable as test aerosols for the PN-PTI application, but special attention should be given to differences that arise from different generator models or set points and consequently for their correction via appropriately defined factors. In view of these results, recommendations were made with regard to PN-PTI type-examination and verification.

**Author contribution**

All authors designed the experiments. TH, DR and AM carried out the measurement campaigns. TH analysed the data with support from DR. KV prepared the manuscript with contributions from all co-authors.

**Competing interests**

The authors declare no competing interests.

**Acknowledgements**

TH, DR, and KV would like to thank Kevin Auderset and Christian Wälchli (both at METAS) for technical support and useful discussions. AM and BG would like to thank Dominique Lesueur and Andrea Bonamin for technical support.

**Funding**

No external funding was used for this study.

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
