# Peer review of "Influence of soot aerosol properties on the counting efficiency"

_Aerosol Research, 2023_

## Author Comment (AC1)

Response to comment

We would like to thank Dr. O. Bischof for the positive feedback. Please find below the point-to-point response to the questions raised.

- The authors state that "In this study, we challenged six different DC-based PN-PTI instruments…". Please explain why you chose to only investigate DC-based instruments and not complement their performance with CPC-based PN-PTI models such as the MAHLE PMU 400 (respectively the Brainbee PMU-400) or the BEA 090 made by Robert Bosch.

  **Answer**: The six different DC-based PN-PTI instruments investigated in this study were chosen based on purely practical reasons: these instruments have been type-approved in Switzerland and we had access to the model prototypes. Unfortunately, none of our collaborators/partner institutes in Switzerland had a MAHLE PMU 400 instrument to lend us, thus this instrument could not be characterised. However, another CPC-based PN-PTI instrument, namely the NPET 3795 (TSI Inc., USA), was characterised in our previous study (Vasilatou et al. 2023; https://doi.org/10.1016/j.jaerosci.2023.106182). As expected, the plateau counting efficiency of the NPET 3785 did not depend on the morphology or the particle charges of the test aerosol.

  We have amended the text as follows: "We focused on DC-based instruments because we expect a larger impact of the aerosol properties on their response compared to CPC-based ones (Vasilatou 2023)".

- It was surprising to see that in case of the MISG, only particles with a GMD of 100 nm were generated, so only one particle size distribution. Have you tried to operate it with a different air/fuel ratio (AFR) or with a fuel other than the mixture of dimethyl ether and propane? It seems odd to see only one data point for the MISG, e.g. in Fig. 3. Note that Bischof et al. (2019) have used the MISG as a calibration aerosol source to determine the counting efficiency of CPCs with DMA-selected particles down to the small nanoparticle size range.

  **Answer:** Actually, our results are in agreement with those reported by Bischof et al. (2020). The modal diameter in their study was 95-158 nm. The concentration achieved at small sizes (<2000 $cm^{-3}$) is not sufficient for DC-based instruments.

  Moreover, our previous study has shown that the counting efficiency of the PN-PTI instruments can change depending on whether the test aerosols are poly- or monodisperse (Vasilatou et al. 2023; https://doi.org/10.1016/j.jaerosci.2023.106182; Figure 6). We therefore chose to challenge the PN-PTI instruments with polydisperse soot aerosols to simulate the GSD of the soot size distribution emitted by diesel vehicles. We have tried to operate the MISG with different fuels and our findings confirm those by Senaratne et al. 2023 (https://doi.org/10.1016/j.jaerosci.2023.106144), who showed that the smallest $GMD_{mob}$ (about 95 nm) is produced when using a mixture of dimethyl ether and propane.

  We have amended the text as follows: In the case of the MISG, particles with a $GMD_{mob}$ down to 100 nm were produced in a repeatable and stable manner using a mixture of dimethyl ether and propane (Senaratne et al., 2023). "This is in agreement with another study, where the modal diameter varied between 95 and 158 nm (Bischof et al. 2020)."

- You state that mobility particle size distributions were measured simultaneously by SMPS, but you do not show any. A particle size distribution of the test aerosol would be interesting to see, e.g. to show the full size range, its concentrations as well as its mode. As is, we can only assume

the aerosol was mono-modal and did not have a nucleation mode (which PN-PTI instruments might experience in real life tests).

**Answer:** Thank you for this comment. Mobility size distributions are shown in Fig. S1 of the Supplement. We have now added the following sentence to the main manuscript (Line 115) to catch the attention of the readers: "Mobility size distributions are shown in Fig. S1."

- The CAP 3070 instrument is based on the so-called escaping current principle, so it detects the current leaving its sensor on charged particles rather than measuring the total net charge carried by particles after collection on a diffusion screen as most DC-sensors do; see Lehtimäki (1983). Could this difference in measurement principle also be a reason for the larger difference in the counting efficiency observed in chapter 3.2, in addition to "an overestimated internal correction factor"?

**Answer:** This is a good point. We cannot rule out that the decrease of the counting efficiency at larger particle sizes is not due to the measurement principle of the device. We have therefore amended the main manuscript (Lines 153-154) as follows: "It cannot be ruled out that the measurement principle of the instrument, based on the so-called escaping current principle, plays also a role (Lehtimäki, 1983)."

---

## Author Comment (AC2)

We would like to thank Dr. Fierz for his comments. Please find below the point-to-point response to the questions raised.

1) The influence of particle morphology on diffusion charging (unipolar and bipolar) is well known, so it is obvious that instruments based on diffusion charging will - at least potentially - be sensitive to particle morphology (you should add one or more references to this).

   **Answer:** We have amended the text as follows (Lines 161-163): This indicates that the exact morphology (e.g. primary particle size, effective density) of the test aerosol had an effect on instrument performance as expected from previous studies (see Dhaniyala et al., 2011 and references therein).

2) To my knowledge, the fractal dimension of the particles is usually quoted when discussing effects on diffusion charging (rather than the density - because the density depends on the particle diameter). It would therefore be better to give an estimate of the fractal dimension of the particles rather than the density, also for comparison with previous studies on particle charging.

   **Answer:** We have now calculated the fractal dimension (see revised Table 1). We have also amended the text as follows: "The fractal dimension $D_f$ of soot particles with a nominal $GMD_{mob}$ of 100 nm was derived via image analysis of high-quality TEM-images using the FracLac feature of ImageJ 1.53e (ImageJ, National institutes of Health, USA). In a first step, the greyscale TEM-images were converted into binary images utilizing the auto-convert function of FracLac. In a second step, the $D_f$ values were determined via the so-called box counting, averaging 12 rotations of each image. The $D_f$ values summarised in Table 1 represent the average values obtained from at least 10 particles for each type of soot. These values agree well with those reported in previous studies for bare (i.e. freshly emitted) soot particles (Pang et al., 2022; Wang et al., 2017)".

3) Also, it is known from the literature that the differences regarding charging due to morphology increase with increasing particle diameter. Since the regulations include counting efficiency limits at 200 nm, where the issue of particle morphology is probably even more critical than at 100 nm, it would have been nice to extend the measurements up to 200 nm. Did you just decide not to measure larger particles, or were there issues of generating a sufficient number of such large particles with your soot generators (just like it is difficult to generate enough smaller particles with the MISG)?

   **Answer:** It is true that the observed deviations in the counting efficiency of the PN-PTI instruments are expected to increase at larger particle sizes. We chose, however, to focus on particles with a $GMD_{mob}$ of up to about 100 nm because these particles are more representative of soot emitted from diesel vehicles (typical GMD mob between 50 and 90 nm). We estimate that soot particles with a mobility diameter of 200 nm would correspond to less than 10 % of the total number of particles emitted by the diesel vehicles, thus their influence on the counting efficiency of the PN-PTI instrument would be small.  For this reason, some countries, e.g. the Netherlands and Belgium do not pose any requirements on the counting efficiency of PN-PTI instruments at a mobility diameter of 200 nm (technical specifications are only defined for particles with a mobility diameter of ≤80 nm).

   We have added the following clarification in Section 1 "Introduction":

   "PN-PTI instruments go through a type-examination procedure which may differ in each country. Among several tests, type-examination includes a counting efficiency and a linearity check

typically performed with combustion aerosols. During their lifetime, PN-PTI instruments are checked for their linearity with polydisperse particles (typically with a $GMD_{mob}$ of 70 ± 20 nm).

…

The geometric mean diameter of the test aerosol was in the range used for linearity checks of PN-PTI instruments as well as in typical size range emitted by diesel engines. The scope of our study was to investigate possible differences that may arise when using different combustion aerosol generators during the type-examination and verification of PN-PTI instruments as well as to correlate with diesel engine emitted soot. We focused on DC-based instruments because we expect a larger impact of the aerosol properties on their response compared to CPC-based ones (Vasilatou et al., 2023)."

4) Pure diffusion charging is unsuitable for particle counting, as larger particles acquire more charge than smaller ones. There are multiple different ways to achieve a more uniform counting efficiency with a diffusion charger so that it can fulfil the PN-PTI specifications. The method chosen to achieve this is crucial for understanding the behavior of the instrument in experiments like this - e.g. how the device will react to precharged particles of either polarity, and also how it will react to the higher/lower charge that particles with higher or lower fractal dimensions  acquire. Therefore, you should briefly explain the operation principle of the different devices. For example, the HEPaC and the DiTEST device use the same principle of operation, so it can be expected that they react in a similar way, whereas at least one of the other 4 devices uses a very different principle of operation and can be expected to react in a different way. Grouping the instruments by principle of operation might help explain the different types of instrument responses you observed.

**Answer:** Dr. Fierz is absolutely right that a discussion on the design of the PN-PTI instruments could be useful in explaining some of the observed differences in their performance. However, most of the manufacturers do not provide any information on the operation principle of their DC-sensor neither on their website nor in the manual. The only relevant information that we could find were publications by Dr. Fierz on the design of the Partector. We would like to urge other instrument manufacturers to openly discuss the design of the DC-based devices with the scientific community and end-users as well.

---

## Author Comment (AC3)

Response to comment

We would like to thank Reviewer 1 for the comments and corrections, which have helped us improve the quality of the manuscript. Please find below the point-to-point response to the questions raised.

1. While the paper presents the counting efficiency results (in different ways), it falls short of discussing in enough depth the "effect" of soot aerosols on the counting efficiency, as the title suggests, and "why" those effects are observed.

**Answer**: The Reviewer is right that an in-depth discussion on why the effects are observed is missing. The reason is that most manufacturers have not disclosed publicly any information on the design of their instrument. It is therefore impossible for us to link the observed measurements (i.e. counting efficiency) with the exact operation principle of each DC-based sensor.

2. It is not clear why the studied particle properties (e.g., EC/TC ratio or primary particle size) would affect the instrument counting efficiency. I understand that diffusion charging depends on the size of the particle and its morphology (if it is not spherical), but particle effective density (at one size), primary particle diameter, and EC/TC ratio do not provide meaningful insight into particle morphology.

**Answer**: We would argue that the EC/TC ratio can have an effect on the morphology of the soot particles. Soot particles formed in premixed flames (i.e. high EC/TC) exhibit a loose agglomerate structure where the primary particles are clearly distinguishable from one another, while soot generated in fuel-rich flames (high OC/TC) has a more compact structure and the primary particles tend to merge with each other (see Fig. 3 in Ess at al. https://doi.org/10.1080/02786826.2021.1901847).

We have amended the text in Subsection 3.1 as follows: "Particle number concentration measured by diffusion chargers depends on the average number of charges carried by each particle (Fierz et al., 2011). Particle size and morphology have been shown to have an effect on the number of charges carried by the particles and, thus, on the counting efficiency of diffusion charger based PN-PTI instruments (see (Dhaniyala et al., 2011; Vasilatou et al., 2023) and references therein). Soot particles form complex structures described by a fractal-like scaling law (Mandelbrot, 1982), and their mobility is influenced by their morphology (described by the fractal dimension and fractal pre-factor) and the momentum-transfer regime (Filippov et al., 2000; Melas et al., 2014; Sorensen, 2011). To characterise the soot particles produced by the different aerosol generators, the following aerosol properties were determined: particle size distribution, EC/TC ratio, primary particle size and fractal dimension. EC/TC ratio can also have an effect on the morphology of the soot particles. Soot particles formed in premixed flames (i.e. high EC/TC) exhibit a loose agglomerate structure where the primary particles are clearly distinguishable from one another, while soot generated in fuel-rich flames (high OC/TC) has a more compact structure and the primary particles tend to merge with each other (see Fig. 3 in (Ess et al., 2021b))."

**Specific comments**

• Why did the authors choose TSI NPET 3795 as the reference particle counter? The authors state that NPET was calibrated according to ISO 27891, so why didn't they use the same reference instrument used to calibrate NPET as the reference particle counter? According to the specifications of NPET, it has a counting efficiency of < 50% at 23 nm and > 50% at 41 nm. The relatively low detection efficiency of NPET at 23 nm and even at 41 nm can potentially lead to unknown counting efficiency of PTI instruments, as some of the (smaller) soot aerosols may not be measured by NPET.

Since this study is done by METAS, I suggest a reference CPC (with a lower d50 size) and a diluter with a known dilution factor be used as the reference particle counter.

**Answer:** This is a good point. We chose to use the NPET as a reference particle counter for the field campaign because it is a robust, portable instrument which combines a stable, reproducible aerosol dilution system and a CPC in a single unit. The NPET was calibrated in the laboratory against a CPC (with a cut-off at around 6 nm) which was equipped with a custom-made dilution system. This system is not designed for field measurement as it is difficult to transport and would require recalibration every day. In our experience, there are no commercial dilution units with a known dilution factor, i.e. the dilution factor does not necessarily remain stable when the instrument is moved from the lab to the field.

We have amended Section 2 as follows: "The NPET was selected as reference instrument for two reasons; i) it could be used in field measurements as it included a dilution system, a volatile particle remover and a particle counter, ii) during type examination portable PN-PTI instruments are typically used as reference".

During data analysis, the particle number concentration reported by the NPET was corrected with respect to the size-dependent counting efficiency, therefore we are confident that the data reported in this study are reliable. We have amended the text as follows: " NPET had been calibrated in a traceable manner according to the ISO 27891 standard, and showed a CE of 0.58 ± 0.02, 0.77 ± 0.02, 0.77 ± 0.01, 0.80 ± 0.01 and 0.79 ± 0.02 at a $GMD_{mob}$ of 23 nm, 50 nm, 70 nm, 80 nm and 100 nm, respectively, and this counting efficiency was taken into account during data analysis".

- The authors have shown TEM images of different soot aerosols, which provide qualitative insight about particle morphology. For example, it is clear from these images that soot from MISG has a very compact structure, while soot from other sources are fractal aggregates. However, it is important to quantify the morphology of particles too to allow studying its effect on counting efficiency. This is typically done by determining the fractal dimension (e.g., through image analysis) or mass-mobility exponent (e.g., measuring particle mass or effective density over a range of particle sizes). The use of effective density at one particle size (100 nm) or primary particle diameter cannot give meaningful information about particle morphology.

**Answer:** Thank you for this comment. We have now calculated the fractal dimension and amended Table 1 (see last column).

We have also amended the text as follows: "The fractal dimension $D_f$ of soot particles with a nominal $GMD_{mob}$ of 100 nm was derived via image analysis of high-quality TEM-images using the FracLac feature of ImageJ 1.53e (ImageJ, National institutes of Health, USA). In a first step, the greyscale TEM-images were converted into binary images utilizing the auto-convert function of FracLac. In a second step, the $D_f$ values were determined via the so-called box counting, averaging 12 rotations of each image. The $D_f$ values summarised in Table 1 represent the average values obtained from at least 10 particles for each type of soot. These values agree well with those reported in previous studies for bare (i.e. freshly emitted) soot particles (Pang et al., 2022; Wang et al., 2017)".

- The recommendations given in Section 4 do not seem to be directly drawn based on the results of this study. Rather, some of the recommendations are generic and seem to be based on the results of other or previous studies.

**Answer:** Recommendations based in previous publications (https://doi.org/10.1016/j.jaerosci.2023.106182) were amended to account for the new findings of this study. For instance, Recommendation 1) highlights that the **same type** of combustion generator

should be used for the determination of CE during type-examination and verification. Recommendation 2) is new and Recommendation 3) highlights the need to verify that type-approval of PN-PTI instruments is harmonised in Europe. Up to now, it was believed that DC-sensors would respond similarly to all types of soot irrespective of the combustion generator. We now show that this not the case, thus type-examination procedures in different European countries might not be equivalent since the approval authorities use different types of soot generators.

- Figures S6 – S9 are not referenced in the main text of the paper.

**Answer:** Thank you for bringing this to our attention. We have amended subsection 3.2 with the following sentence: "The counting efficiency of the different PN-PTI counters as a function of time is shown in Figs. S6-S9 for a measurement duration of 2 min".

- Table S1: It seems that EC/TC ratio is typed mistakenly as "EC/OC ratio". In any case, EC/TC ratio is also given in Table 1, so I suggest providing this information in one place only (either in Table 1 or S1).

**Answer:** Thank you for spotting this typo. It now reads "EC/TC mass fraction (%)". Although the EC/TC mass fraction is provided in the main manuscript, we wanted to provide in SI a comprehensive summary of the setpoints we used during the study, so that the readers can find all relevant information in one Table.

---

## Author Comment (AC4)

Response to Review

We would like to thank Reviewer 2 for the constructive feedback. Please find below the point-to-point response to the questions raised.

- The paper examines the properties of soot that influence the counting efficiency of instruments for periodic technical inspection. The instruments used in the study are all based on diffusion charging. Since diffusion charging by itself is particle size dependent, different operating strategies are used to limit the influence of particle size. Therefore, the operating principles of the instruments differ significantly. A comprehensive description of the measurement principles used and explanations for the different responses to soot based on the measurement principle would be a very valuable contribution.

**Answer**: This is a good point but, unfortunately, most manufacturers have not officially revealed any information on the exact operating principle of their DC-sensor. The only relevant publications we could find were by Dr. Martin Fierz on the Partector. We are very grateful to all manufacturers who lent us instruments for this study, especially for giving us their consent to publish the results naming explicitly the devices (many studies in the past have only published anonymised results), and we would like to encourage them to share more information on the design of their DC sensors with the aerosol community.

- As the size dependence is crucial, it would have been very interesting to see larger sizes (e.g. up to 200 nm) as some legislation requires CEs up to 200 nm.

**Answer**: We chose to focus on particles with a $GMD_{mob}$ of up to about 100 nm because these particles are more representative of soot emitted from diesel vehicles (typical GMD mob between 50 and 90 nm). We estimate that soot particles with a mobility diameter of 200 nm would correspond to less than 10 % of the total number of particles emitted by the diesel vehicles, thus their influence on the counting efficiency of the PN-PTI instrument would be small. For this reason, some countries, e.g. the Netherlands and Belgium do not pose any requirements on the counting efficiency of PN-PTI instruments at a mobility diameter of 200 nm (technical specifications are only defined for particles with a mobility diameter of ≤80 nm).

We have added the following clarification in Section 1 "Introduction":

"PN-PTI instruments go through a type-examination procedure which may differ in each country. Among several tests, type-examination includes a counting efficiency and a linearity check typically performed with combustion aerosols. During their lifetime, PN-PTI instruments are checked for their linearity with polydisperse particles (typically with a $GMD_{mob}$ of 70 ± 20 nm).

…

The geometric mean diameter of the test aerosol was in the range used for linearity checks of PN-PTI instruments as well as in typical size range emitted by diesel engines. The scope of our study was to investigate possible differences that may arise when using different combustion aerosol generators during the type-examination and verification of PN-PTI instruments as well as to correlate with diesel engine emitted soot. We focused on DC-based instruments because we expect a larger impact of the aerosol properties on their response compared to CPC-based ones (Vasilatou et al., 2023)."

- It is well known that (diffusion) charge-based sensor principles are sensitive to particle morphology. A discussion of this influence as well as a morphological characterisation (e.g. fractal dimension) of the soot produced by the different generators is missing.

**Answer**: Thank you for this comment. We have now calculated the fractal dimension and amended Table 1 (see last column).

We have also amended the text as follows: "The fractal dimension $D_f$ of soot particles with a nominal $GMD_{mob}$ of 100 nm was derived via image analysis of high-quality TEM-images using the FracLac feature of ImageJ 1.53e (ImageJ, National institutes of Health, USA). In a first step, the greyscale TEM-images were converted into binary images utilizing the auto-convert function of FracLac. In a second step, the $D_f$ values were determined via the so-called box counting, averaging 12 rotations of each image. The $D_f$ values summarised in Table 1 represent the average values obtained from at least 10 particles for each type of soot. These values agree well with those reported in previous studies for bare (i.e. freshly emitted) soot particles (Pang et al., 2022; Wang et al., 2017)".

-   The rationale for the selection of particle properties (EC/TC mass fraction, effective density, primary particle size) to assess the counting characteristics of the PN-PTI instruments is unclear and should be explained.

**Answer**: We have amended the text in Subsection 3.1 as follows: "To characterise the aerosols, the following aerosol properties were determined: particle size distribution, EC/TC ratio, primary particle size and fractal dimension. Particle size and morphology have been shown to have an effect on the counting efficiency of PN-PTI instruments (see (Vasilatou et al., 2023) and references therein). EC/TC ratio can also have an effect on the morphology of the soot particles. Soot particles formed in premixed flames (i.e. high EC/TC) exhibit a loose agglomerate structure where the primary particles are clearly distinguishable from one another, while soot generated in fuel-rich flames (high OC/TC) has a more compact structure and the primary particles tend to merge with each other (see Fig. 3 in Ess at al. https://doi.org/10.1080/02786826.2021.1901847)."

-   Chapter 4 makes recommendations based on the results of the study. What specific results led to the recommendations? Why is soot the best calibration aerosol? It seems that BigCAST Aerosol gives different CE results (e.g. 1.4 in Fig. 5a and 1.0 in Fig. 5b for CAP3070).

**Answer**: In our opinion, combustion-based soot is the most suitable calibration aerosol because it reproduces well the morphology and size of soot from diesel engines. As shown in our previous study, particles from spark discharge generators and salt nebulisers produce particles that are too fractal or have a cubic shape, respectively, and thus lead to deviations up to a factor of 2 in the CE efficiency of PN-PTI instruments with respect to soot from a Euro 3 diesel engine (see, for instance, Figure 5 in Vasilatou et al. https://doi.org/10.1016/j.jaerosci.2023.106182). The deviations in CE when using combustion-based aerosol with respect to a Euro 5b engine are much smaller (see Figure 4a of present study).

-   Minor: Why is a rather complicated aerosol aftertreatment (CS, dehumidifier, 1:10 diluter, blower, dilution bridge, …) after the soot generators used? Why is it different for the EU5b exhaust?

**Answer**: The aerosol aftertreatment between lab experiments (Fig. 5a) and field measurements (Fig. 5b) is quite similar in our opinion. In both cases there is a catalytic stripper, a dilution unit and a custom-made dilution bridge.

---

## Author Response (AR1)

A separate response has been uploaded for each review and Comment posted.

---

## Referee Report (RR1)

**Referee Report on ar-2023-16**

Title: Influence of soot aerosol properties on the counting efficiency of PN-PTI instruments

Authors: Tobias Hammer, Diana Roos, Barouch Giechaskiel, Anastasios Melas, and Konstantina Vasilatou

**General Comments**

The paper describes a follow-up study of Vasilatou et al. 2023, focusing on the influence of various polydisperse soot aerosols for the in-field calibration of DC based PN counters for PTI of Diesel vehicles. It reveals that the aerosol properties affect the CE of the instruments under test. In the actual study, the authors focused on combustion generated soot particles. In the previous study, the influence on the CE was enormous when using various types of test aerosols (salt, soot-like). In this study, when using combustion generated soot particles only, the influence on the CE is still significant, but could not fully be explained by the particle morphology, that was claimed the main influence quantity besides the particle size. Based on the findings, the authors recommend the usage of one of the used soot generators for type approval and annual calibration.

**Strengths of the paper**

- This work revealed the importance of test aerosol properties, like the particle material, which was not addressed in the various device specifications for PN PTI measurement.
- The experimental methods were well-founded (except the choice of the reference instrument), and the properties of the test aerosols were well characterized.

**Weaknesses of the paper**

- The main flaw of the paper is, that it does not deliver an explanation of the influence mechanisms on the CE of the various instruments. There is absolutely no common trend towards the various instruments, therefore, it is not possible to draw general conclusions about DC based instruments or the usage of generators. The aerosol properties were investigated regarding the particle morphology quite detailed. At the beginning of Section 3.1 it is stated, that DC based instruments are generally affected by the particle morphology because the average number of charges within the diffusion charging process will vary with the particle shape. However, there is no deeper explanation on how the instruments' CE is affected by morphology. Other studies or explanation by the manufacturers should be considered for the paper!
- The second point is, that the choice of the reference instrument is questionable, because it is just one PTI instrument and the bias of -23% against the METAS reference could be problematic for practical applications. There are at least two questions that need to be answered.
  1. Was the reference instrument tested against other METAS references up to the required concentrations of 5E6 cm$^{-3}$ using polydisperse particles?? (The ISO27891 does not include a linearity measurement with polydisperse aerosols!)
  2. An explanation why the CE is up to -23% against the METAS reference must be given! Was there a traceable calibration by the manufacturer before? Why is

there such a large difference? Was the instrument used before calibration and did it drift to such an extend??

If this instrument would be used as a reference in the field, a bias of -23% could be problematic since the maximum permissible error in some regulations (like the NPTI proposal) is 25% only!!! The *"COMMISSION RECOMMENDATION of 20.3.2023 on particle number measurement for the periodic technical inspection of vehicles equipped with compression ignition engines"* also states the requirement that the reference systems' MU shall be less than 20% for subsequent verification!

- There was no added value from the profound investigation of the test aerosol properties. None of the parameters listed in table 1 delivered an explanation of the CE behavior of the instruments under test.
- Is the test aerosol from one EURO5 vehicle representative for "Diesel soot" in general?

**Specific comments**

- Line 93: reference instrument
    1. Was this instrument tested against other METAS references up to the required concentrations of $5E6$ cm$^{-3}$?? (The ISO27891 does not include a linearity measurement with polydisperse aerosol)
    2. Please explain why CE is -23% against the METAS reference! Was there a traceable calibration by the manufacturer before? Why is there such a large difference? Was the instrument used before calibration and did it drift to such an extend??

    If this instrument would be used as a reference in the field, a bias of -23% could be problematic, since the maximum permissible error in some regulations (like the NPTI proposal) is 25% only!!! The " COMMISSION RECOMMENDATION of 20.3.2023 on particle number measurement for the periodic technical inspection of vehicles equipped with compression ignition engines" also states the requirement that the reference systems MU shall be less than 20% for subsequent verification!

- Line 123: aerosol properties

    The investigation of the aerosol properties was interesting and detailed but limited to the morphology of the particles. This is fine under the assumption that the particle morphology is the main parameter that affects the CE of DC based instruments. Are you sure that there is no other parameter, that affects the found behavior in the same order of magnitude?

- Line 189/190 & Line 195/196

    It would be interesting to have an explanation how the CE varies with different GMDs and particle materials. To my impression, this very much depends on the measuring principle of each individual device, and thus, it would be good to explain the differences by the various measuring principles!

- Line 202/203

    This conclusion is questionable! An explanation of the scattering of the values must be delivered, rather than conducting the test at more comfortable test conditions!!

- Figure 5b)

    The comparison with Vasilatou et al., 2023 shows, that there might be other influence quantities than the particle morphology only. The importance of pre-existing charges is obvious in figure 5b) and might be even more important if there are internal correction

factors used, as in the case of CAP3070! I recommend to read Knoll et al., 2021 regarding influence of pre-charges on DC based instruments.

- Line 297/298
  The link to this reference does not work!
- Line 204 / Figures S6 – S9 in supplement
  There were some significant fluctuations of the CE signal using the differnet aerosol generators:
  1. The DX280 and the Knestel had big fluctualtions with the MiniCast 5201 under fuel rich conditions
  2. The fluctualtions of all instruments, except the HEPaC and the CAP3070 were very large with the MiniCAST 6204
  3. The AEM had fluctuations of about 50% CE using the MISG without cyclone

  --> Please explain why those signals were so unstable!!!

  --> What was the reason (pressure fluctuations, flow rate, heating, signal processing of instruments, etc.) and why is there no clear trend towards the instruments and the generators used?

  --> Can you exclude instabilities of the reference instrument?

**Further literature to be considered**

- Bainschab et al., 2020 (https://doi.org/10.1016/j.aeaoa.2020.100095)
  Calculation of false pass / false fail scenarios and general impact of PN PTI on fleet emissions.
- Krasa et al., 2023
  https://tandf.figshare.com/articles/journal_contribution/Toward_a_simplified_calibration_method_for_23_nm_automotive_particle_counters_using_atomized_inorganic_salt_particles/22121581
  For the sake of completeness in the introduction, the study of Krasa et al., 2021 should be mentioned. It shows a significant impact of the test aserosol also for CPCs.
- Knoll et al., 2021 (https://doi.org/10.1080/02786826.2021.1873910)
  Influence of pre-charges on DC based instruments.

---

## Author Response (AR2)

**We would like to thank all Reviewers for their valuable feedback. Please find below the point-by-point response to the questions raised.**

**Referee Report 1**

This study compares the performance of a range of particle counters for periodic technical inspection on different polydisperse aerosols, such as soot produced by combustion. With its methodology, content and conclusion it is an extension to a previous study done by Vasilatou et al (2023) "Effects of the test aerosol on the performance of periodic technical inspection particle counters". This review is written based on the track changed version/ reviewed first manuscript.

**General Comments / Questions:**

- Line 241: "… test aerosols such as NaCl…" – Figure 5 a and b show great deviation for all types of test aerosols and instruments. Figure 5a up to 50%. Nebulised NaCl tends to have a large variety of GMD (Global Mean Diameter), depending on its concentration of the nebulised Liquid (Liu and Lee, 1975). Since the counting efficiency of diffusion chargers depends on the size distribution (Fierz et al 2007), it is possible to have a fitting salt GMD as well. Vasilatou et al 2023 examined just one possible NaCl GMD.

**Answer**: We respectfully disagree with this comment. Vasilatou et al. (2023) examined four different salt generators (FCS 249, ATM 228, ATM 220 and Meinhard) and various $GMD_{mob}$ in the range 30 - 120 nm (see Figs. 3, 5 and 7 of the respective publication). In the manuscript under review we focus, however, on test aerosols with a GDM of 80 nm as required by the Swiss, Dutch and Belgian legislation for the yearly PN-PTI verification (in Germany the requirement is GDM = 70 ± 20 nm).

- A Previous Reviewer (https://doi.org/10.5194/ar-2023-16-RC1) posted: "The recommendations given in Section 4 do not seem to be directly drawn based on the results of this study. Rather, some of the recommendations are generic and seem to be based on the results of other or previous studies" I would agree upon this statement! Please clarify this in the text, that the recommendations are done (Line 250) based on the results of this study and previous.

**Answer**: We have amended the sentence as follows: "Based on the results of this and previous studies (Vasilatou et al., 2023), the following recommendations can be made…".

- Recommendations key message:

-Recommendation 1: Vasilatou et al (2023): "…highly recommended to use particles for calibration purposes which are similar in chemistry and morphology to soot particles emitted by vehicle engines" and Hammer et al. (2023): PN-PTI counters should ideally be performed with soot as test aerosol"

Recommendation 2 is about the setup correction factors for both. The Conclusions are short with limited discussions. Line 281: "This study confirms that soot aerosols …. Are more suitable than NaCl". The only data for this statement are taken from Vasilatou et al 2023! Even there the NaCl data is limited. No comparison for different NaCl GMD. And NaCl data is within the same range as Soot aerosol.

**Answer**: As mentioned previously, Vasilatou et al. examined four different nebulisers (FCS 249, ATM 228, ATM 220 and Meinhard) and various $GMD_{mob}$ in the range 30 - 120 nm (see Figs. 3, 5 and 7 of the respective publication). We do not agree that the data are limited. The size distribution of NaCl aerosols **must** be within the same range as for the soot test aerosols. National legislations prescribe specific $GMD_{mob}$ ranges for instrument type-examination and verification. Both studies were carried out in the particle size range defined by national regulations.

- Missed opportunity for a DMA -CPC to DMA - PN-PTI cut-off efficiency experiment. This study would benefit from this insight.

**Answer**: We thank the reviewer for this comment. Our scope was to examine the verification of PN-PTI instruments, which is exclusively carried out with polydisperse particles. Monodisperse soot aerosols are sometimes used for instrument type-approval, but this was out of the scope of the current study.

- Please add grid lines for all figures

**Answer**: Certain journals recommend not to use grid lines in the figures. If the journal editor advises is to add grid lines, we will be happy to do so.

Specific Comments:

- Line 17 – Abstract: … "0.25 units" … - do you mean "%" , in that case, Figure 3 and Figure 4 shows that this threshold was exceeded many times.

**Answer**: The ideal counting efficiency of an instrument is 1 (= 100%). 0.25 units refers to the counting efficiency of 1. This would be equal to 25 % if the counting efficiencies were expressed as 100%.

- Line 21 – Abstract: …"MISG may be satisfactory" – that conclusion is based on the instrument response and ET/TC mass fraction, but effective density and particle size are in clearly different. Please clarify.

**Answer**: This is clarified is Section 4: "Low-cost soot generators can be a stable source of combustion particles and can be employed for PN-PTI verification using the appropriate setup correction factors. However, the GMD they produce should be in the range 70±20 nm in order to comply with the current linearity verification requirements in Europe". Note that national legislations do not specify a range for the effective particle density of the test aerosols.

- Line 89 : "… 8 port flow splitter" – Are concentration adjustments based on line loss (towards the CPC) or flow/pathlength to each individual instrument be considered?

**Answer**: Thank you for bringing this up. We have added the following sentence to explain how we ensured that the losses are equal: "The length of the tubes from the flow splitter to the devices was adapted to the respective flow rate to ensure equal diffusion losses".

- Line 95 + 185: "… counting efficiency was taken into account" – As you mention it multiply times, I would assume, that you multiply the Particle Number Concentration with an correction factor, please clarify in the figures as well.

**Answer**: We would argue that the most suitable place to describe how the measurements were performed is in the Section "Materials and Methods" as we have already done.

- Line 204: "… duration of 2 min" – Why was 2 minutes chosen? Duration for a standard engine exhaust test?

**Answer**: Yes, this is explained in Lines 120-121: "… thus the duration was similar to the duration of real PN-PTI tests which varies from 15 to 90 s".

- Line 231: "… tends to be larger …"- nearly a factor of 2! Please clarify.

**Answer**: In Lines 265-267 we highlight the fact that "the properties of real diesel soot can also differ considerably, depending on the engine design, driving cycle and fuel properties (Hays et al., 2017; Wihersaari et al., 2020)". This is why PN-PTI instruments are type-examined in the particle size range 23 nm – 200 nm. Aerosols with a $GMD_{mob}$ of 100 nm generated by the MISG fall within this size range and are very close to the recommended size range of 70 ± 20 nm for PN-PTI verifications.

- Line 256: "range of 70+-20 nm" you mention in the Abstract, that the MISG is suitable, but in your recommendations it wouldn't fit. Please Clarify

**Answer**: In the abstract, we state that "MISG **may** be a satisfactory - and affordable - option for PN-PTI verification". In Section 4 (Recommendations) we further explain that the MISG would need to fulfil the requirement 70 ± 20 nm to comply with the current linearity verification requirements in Europe. This might be possible in the near future by using a different fuel mixture that has not been tested so far or by stabilising the various flows to make generation of smaller particle size distributions more reproducible. To avoid any misunderstanding, we have amended the Abstract as follows: "… however, further optimization will be needed for low-cost soot generators to comply with European PN-PTI verification requirements".

References

Vasilatou, K., Wälchli, C., Auderset, K., Burtscher, H., Hammer, T., Giechaskiel, B. and Melas, A.: Effects of the test aerosol on the performance of periodic technical inspection particle counters, J. Aerosol Sci., 172(January), doi:10.1016/j.jaerosci.2023.106182, 2023.

Liu, B.Y.H., and K.W. Lee (1975) "An Aerosol Generator of High Stability" Am. Ind. Hyg. Assoc. J., 36, 861–865

Fierz, M., Burtscher, H., Steigmeier, P., and Kasper, M., "Field Measurement of Particle Size and Number Concentration with the Diffusion Size Classifier (Disc)," SAE Technical Paper 2008-01-1179, 2008, https://doi.org/10.4271/2008-01-1179.

**Referee Report 2**

General comments

This manuscript describes the relative response of PN-PTI instruments with a variety of soot sources including one Euro 5b diesel vehicle. Such a study is needed because non-soot sources (NaCl and spark-discharge) have been shown to produce instrument responses that greatly differ from soot. This study shows that a wide range of soot sources could be used to meet the specification requirements for PN-PTI calibration, and thus, this study should be published as it will help develop standards and legislation.

Specific Comments

- Line 85. Figure 1 shows a blower but it is not described in the text. Please state why it was used.

**Answer**: Thank you for pointing this out. We have amended the text as follows: "To deliver the aerosol into the mixing volume, a blower (Micronel AG, Switzerland) was used".

- Line 85. Figure shows that compressed air was added to the mixing chamber but it was not described in the text.

**Answer:** We have amended the text as follows: "…a custom-made dilution bridge, and was mixed and diluted with filtered air in a 27-ml-volume chamber".

- Line 85. It is a good idea to use a mixing volume as was done. However, verification that the aerosol is well mixed is after the flow splitter is also a good idea. Were checks made to ensure the aerosol was well mixed (i.e. rotating instruments between sampling ports)?

**Answer:** Thank you for bring this to our attention. We have added the following sentence: "The splitter bias was determined according to the procedure specified in the ISO 27891 standard and was found to be within 1 % for particles with a $GMD_{mob}$ equal to or larger than 23 nm".

- Line 151; "The fractal dimension Df of soot particles with a nominal GMDmob of 100 nm was derived via image analysis …." It's not clear how this was done. Was a DMA used to pre-classify the particles onto TEM grids, or did you assume that 100 nm projected area was approximately the equivalent to 100 mobility diameter? Please explain.

**Answer:** We have now repeated the analysis by size-selecting the soot particles with a DMA and Table 1 has been revised accordingly. The calculated Df of size-selected particles are more accurate than the values reported before, but since they still lie in the same range (1.55 – 1.65) the discussion does not change significantly.

We have also modified the text as follows:

Lines 128-129: "The fractal dimension $D_f$ of size-selected soot particles with a mobility diameter $d_p$ of 100 nm was derived via image analysis of high-quality TEM-images using the FracLac feature of ImageJ 1.53e (ImageJ, National institutes of Health, USA)."

Lines 186-188: The calculated fractal dimension of soot particles lied in the range 1.55 – 1.65 for all generators, in line with the fractal-like morphology observed in the TEM images and with previous studies on freshly emitted soot particles from different combustion sources (Pang et al., 2023).

In the case of the Euro 5b diesel vehicle, we could not repeat the measurement with size-selected particles, therefore we deleted the calculated Df value from Table 1.

- Section 3.1. This section is supposed to be for results, however, the authors put many experimental method details which are better placed in Section 2. For example, Lines 151 to 155.

**Answer:** The Reviewer is right. We have moved the text related to the determination of the effective density and fractal dimension into Section 2 "Materials and methods".

- Line 158. The effective density measurement method should be described in Section 2.

**Answer:** We have shifted the sentence: "The effective density was determined for the 100 nm setpoints using an Aerodynamic Aerosol Classifier (AAC, Cambustion, UK) and a DMA (TSI Inc., USA) in tandem as described in (Tavakoli and Olfert, 2014)" to Section 2.

- Line 160. The effective density results could be put into context by comparing them to summary work by Olfert and Rogak (https://doi.org/10.1080/02786826.2019.1577949). They show the 'average' effective of denuded soot is 0.51 g/cm^3 at 100 nm mobility diameter, although there is fair amount of spread in the data and CI engines tend to have higher effective densities.

**Answer**: Thank you for bringing this article to our attention. We have added the following text: "According to the summary work by Olfert and Rogak, the effective density of denuded soot from various sources (gas turbines, compression ignition engines and laboratory-based burners) lies typically in the range 0.4-0.8 g/cm$^3$ at 100 nm mobility diameter (Olfert and Rogak, 2019) Compression-ignition engines tend to produce soot with higher effective densities, while gas-turbine soot tends to have lower effective densities (Olfert and Rogak, 2019)".

**Referee Report 3**

General Comments

The paper describes a follow-up study of Vasilatou et al. 2023, focusing on the influence of various polydisperse soot aerosols for the in-field calibration of DC based PN counters for PTI of Diesel vehicles. It reveals that the aerosol properties affect the CE of the instruments under test. In the actual study, the authors focused on combustion generated soot particles. In the previous study, the influence on the CE was enormous when using various types of test aerosols (salt, soot-like). In this study, when using combustion generated soot particles only, the influence on the CE is still significant, but could not fully be explained by the particle morphology, that was claimed the main influence quantity besides the particle size. Based on the findings, the authors recommend the usage of one of the used soot generators for type approval and annual calibration.

Strengths of the paper

• This work revealed the importance of test aerosol properties, like the particle material, which was not addressed in the various device specifications for PN PTI measurement.

• The experimental methods were well-founded (except the choice of the reference instrument), and the properties of the test aerosols were well characterized.

Weaknesses of the paper

• The main flaw of the paper is, that it does not deliver an explanation of the influence mechanisms on the CE of the various instruments. There is absolutely no common trend towards the various instruments, therefore, it is not possible to draw general conclusions about DC based instruments or the usage of generators. The aerosol properties were investigated regarding the particle morphology quite detailed. At the beginning of Section 3.1 it is stated, that DC based instruments are generally affected by the particle morphology because the average number of charges within the diffusion charging process will vary with the particle shape. However, there is no deeper explanation on how the instruments' CE is affected by morphology. Other studies or explanation by the manufacturers should be considered for the paper!

**Answer**: We have discussed the results with various instrument manufacturers and they could not offer an explanation. In addition, the manufacturers have not published any information on the exact design of their sensors neither in peer-reviewed articles nor in the instruments' manual. Without knowledge of the sensor design, especially the type of charger and voltage applied, it is impossible to interpret the results. We would like to encourage the manufacturers to work together with the aerosol research community and discuss these issues openly, but we can also understand if they do not want to disclose any company secrets.

• The second point is, that the choice of the reference instrument is questionable, because it is just one PTI instrument and the bias of -23% against the METAS reference could be problematic for practical applications. There are at least two questions that need to be answered.

1. Was the reference instrument tested against other METAS references up to the required concentrations of 5E6 cm-3 using polydisperse particles?? (The ISO27891 does not include a linearity measurement with polydisperse aerosols!)

**Answer**: The NPET relies on a CPC sensor and is therefore a very reliable instrument, which yields reproducible measurements. As we have shown in a previous publication (https://doi.org/10.1016/j.jaerosci.2023.106182), the plateau efficiency of the NPET does not depend on the test aerosol material, making it a suitable reference instrument. Furthermore, the NPET was calibrated against the METAS primary standard for number concentration according to the ISO 27891 standard and its counting efficiency was taken into account during data evaluation. The linearity of the NPET was checked and was found to be very close to 1. We see absolutely no reason why the NPET cannot be used as a reference instrument for calibrating DC-based PTI counters in the field.

2. An explanation why the CE is up to -23% against the METAS reference must be given! Was there a traceable calibration by the manufacturer before? Why is there such a large difference? Was the instrument used before calibration and did it drift to such an extend?? If this instrument would be used as a reference in the field, a bias of -23% could be problematic since the maximum permissible error in some regulations (like the NPTI proposal) is 25% only!!! The "COMMISSION RECOMMENDATION of 20.3.2023 on particle number measurement for the periodic technical inspection of vehicles equipped with compression ignition engines" also states the requirement that the reference systems' MU shall be less than 20% for subsequent verification!

**Answer**: We thank the reviewer for this comment. NPET was chosen as reference instrument because in a previous study we found that its plateau efficiency did not depend on the test aerosol material (this information was added in the paper). Before the testing campaign, we calibrated the instrument. The values reported in the paper (deviations of 23 % etc.) were taken into account during the data evaluation and any bias related to NPET counting efficiency was eliminated. In order to avoid confusion, we now report in the manuscript the calibration factor we determined. In addition, the NPET linearity was almost 1 in the concentration range relevant to our study. Moreover, the commission recommendation refers to official PTI tests of diesel engines, not to research studies as the one under review. The goal of our study was to test and compare PTI instruments with different soot aerosols, not to type-approve diesel vehicles.

• There was no added value from the profound investigation of the test aerosol properties. None of the parameters listed in table 1 delivered an explanation of the CE behavior of the instruments under test.

**Answer**: The profound investigation of the test aerosol properties was considered necessary for explaining different CEs of DCs with different soot generators. Unfortunately, this was not possible due to the absence of information on the design of the counters and possible built-in corrections. However,

we strongly believe that a profound investigation is useful for this research field as well as for the ongoing discussion on the design of DC sensors. The conclusions of this study may not give input on DC-sensor optimization but do point out possible sources of errors in PN-PTI and how they can be (at least) reduced.

• Is the test aerosol from one EURO5 vehicle representative for "Diesel soot" in general?

**Answer**: As already noted in the manuscript: "Further studies with more diesel test vehicles would be necessary to elucidate which type of laboratory-generated soot is the best proxy for diesel soot, keeping in mind that the properties of real diesel soot can also differ considerably, depending on the engine design, driving cycle and fuel properties (Hays et al., 2017; Wihersaari et al., 2020)".

• Specific comments - Line 93: reference instrument: Was this instrument tested against other METAS references up to the required concentrations of 5E6 cm-3?? (The ISO27891 does not include a linearity measurement with polydisperse aerosol)

**Answer**: Please see answer above related to the same question.

2. Please explain why CE is -23% against the METAS reference! Was there a traceable calibration by the manufacturer before? Why is there such a large difference? Was the instrument used before calibration and did it drift to such an extend?? If this instrument would be used as a reference in the field, a bias of -23% could be problematic, since the maximum permissible error in some regulations (like the NPTI proposal) is 25% only!!!

The " COMMISSION RECOMMENDATION of 20.3.2023 on particle number measurement for the periodic technical inspection of vehicles equipped with compression ignition engines" also states the requirement that the reference systems MU shall be less than 20% for subsequent verification!

**Answer**: Please see answer above related to a very similar question.

- Line 123: aerosol properties: The investigation of the aerosol properties was interesting and detailed but limited to the morphology of the particles. This is fine under the assumption that the particle morphology is the main parameter that affects the CE of DC based instruments. Are you sure that there is no other parameter, that affects the found behavior in the same order of magnitude?

**Answer**: We never claimed that particle morphology is the only parameter that affects the counting efficiency of PTI counters. On the opposite, we stated that: "Particle number concentration measured by diffusion chargers depends on the average number of charges carried by each particle (Fierz et al., 2011). Particle size and morphology have been shown to have an effect on the number of charges carried by the particles and, thus, on the counting efficiency of diffusion charger based PN-PTI instruments (see (Dhaniyala et al., 2011; Vasilatou et al., 2023) and references therein). Soot particles form complex structures described by a fractal-like scaling law (Mandelbrot, 1982), and their mobility is influenced by their morphology (described by the fractal dimension and fractal pre-factor) and the momentum-transfer regime (Filippov et al., 2000; Melas et al., 2014; Sorensen, 2011)". In this study, we performed experiments at different particle sizes and we characterised the test particles not only in terms of particle morphology, but also in terms of effective density, EC/TC ratio and fractal dimension.

- Line 189/190 & Line 195/196 It would be interesting to have an explanation how the CE varies with different GMDs and particle materials. To my impression, this very much depends on the measuring principle of each individual device, and thus, it would be good to explain the differences by the various measuring principles!

**Answer**: See answer to similar question above. The exact design and measurement principle of the sensors is not known.

- Line 202/203 This conclusion is questionable! An explanation of the scattering of the values must be delivered, rather than conducting the test at more comfortable test conditions!!

**Answer**: As already mentioned in the manuscript, the performance of DC-based sensors depends on many different parameters (e.g. particle size, morphology, particle charge etc.) and it is impossible to disentangle the effects of each aerosol property on the counting efficiency of the sensors, especially when information on the design of the instruments is not available. But even if one could explain why the PN-PTI instruments show different counting efficiencies, this would not change the fact that the values scatter more at particle sizes larger than 90 nm. Our conclusion that "This supports the choice of soot with 50-90 nm mobility diameter for the PN-PTI instruments verification linearity tests" is therefore valid.

- Figure 5b) The comparison with Vasilatou et al., 2023 shows, that there might be other influence quantities than the particle morphology only. The importance of pre-existing charges is obvious in figure 5b) and might be even more important if there are internal correction factors used, as in the case of CAP3070! I recommend to read Knoll et al., 2021 regarding influence of pre-charges on DC based instruments.

**Answer**: We fully agree that initial particle charge is important. This has been discussed in Vasilatou et al. (https://doi.org/10.1016/j.jaerosci.2023.106182), where the results shown in Fig. 5b are taken from.

- Line 297/298 The link to this reference does not work!

**Answer**: Thank you for bringing this to our attention. It seems that the page has been removed from the EU website. We have revised the reference as follows: Anon: Proposal Particulate Number Counters, [online] Available from: https://nmi.nl/special-particle-number-counters/, n.d.

- Line 204 / Figures S6 – S9 in supplement There were some significant fluctuations of the CE signal using the differnet aerosol generators:

1. The DX280 and the Knestel had big fluctualtions with the MiniCast 5201 under fuel rich conditions
2. The fluctualtions of all instruments, except the HEPaC and the CAP3070 were very large with the MiniCAST 6204 3. The AEM had fluctuations of about 50% CE using the MISG without cyclone --> Please explain why those signals were so unstable!!! --> What was the reason (pressure fluctuations, flow rate, heating, signal processing of instruments, etc.) and why is there no clear trend towards the instruments and the generators used?

**Answer**: The environmental conditions in the laboratory were very stable, so was the aerosol generation. The fluctuations are due to the PN-PTI instruments. Note that these are mid-cost instruments designed with the PTI test in mind, i.e. to check whether diesel vehicles emit above the limit value (e.g. 250'000 cm$^{-3}$). Considering that the reported number concentration is typically averaged over 30-90 sec, the observed fluctuations have little to no influence on the end result, i.e. whether a diesel vehicle will pass or fail the PTI test.

- Can you exclude instabilities of the reference instrument? Further literature to be considered - Bainschab et al., 2020 (https://doi.org/10.1016/j.aeaoa.2020.100095) Calculation of false pass / false fail scenarios and general impact of PN PTI on fleet emissions. - Krasa et al., 2023 https://tandf.figshare.com/articles/journal_contribution/Toward_a_simplified_calibration_ method_for_23_nm_automotive_particle_counters_using_atomized_inorganic_salt_parti cles/22121581 For the sake of completeness in the introduction, the study of Krasa et al., 2021

should be mentioned. It shows a significant impact of the test aserosol also for CPCs. - Knoll et al., 2021 (https://doi.org/10.1080/02786826.2021.1873910)  Influence of pre-charges on DC based instruments.

**Answer**: The NPET has been calibrated multiple times against our primary standard at METAS the past few years and has shown very good stability and reproducibility. Any instabilities of the NPET are negligible compared to the other measurement uncertainties.

Additional correction by the authors: We have spotted an error in Figure 5 and revised it so that the values are in agreement with the measurements published in Vasilatou et al. https://doi.org/10.1016/j.jaerosci.2023.106182.

---

## Author Response (AR3)

We would like to thank the Editor for the valuable feedback.

Lines 100-103: We have amended the sentence as follows: NPET had been calibrated in a traceable manner according to the ISO 27891 standard, and showed a CE of 0.58 ± 0.02, 0.77 ± 0.02, 0.77 ± 0.01, 0.80 ± 0.01 and 0.79 ± 0.02 at a $GMD_{mob}$ of 23 nm, 50 nm, 70 nm, 80 nm and 100 nm, respectively and this counting efficiency was taken into account during data analysis (i.e. calibration factors in the range 1.72 - 1.28 were applied to the concentrations reported by the NPET depending on the particle size).

Line 98: We now explain the abbreviation CS as follows: "… a catalytic stripper (CS, Catalytic Instruments GmbH, Germany)…".